# Comprehensive blueprint of *Salmonella* genomic plasticity identifies hotspots for pathogenicity genes

Simran Krishnakant Kushwaha[1,2], Yi Wu[2], Hugo Leonardo Avila[3], Abhirath Anand[4], Thomas Sicheritz-Pontén[5,6], Andrew Millard[7], Sandhya Amol Marathe[1], Franklin L. Nobrega[2]*

1 Department of Biological Sciences, Birla Institute of Technology & Science (BITS), Pilani, Rajasthan, India, 2 School of Biological Sciences, University of Southampton, Southampton, United Kingdom, 3 Laboratory for Applied Science and Technology in Health, Instituto Carlos Chagas, FIOCRUZ Paraná, Brazil, 4 Department of Computer Sciences and Information Systems, Birla Institute of Technology & Science (BITS), Pilani, Rajasthan, India, 5 Center for Evolutionary Hologenomics, Globe Institute, University of Copenhagen, Copenhagen, Denmark, 6 Centre of Excellence for Omics-Driven Computational Biodiscovery (COMBio), AIMST University, Bedong, Kedah, Malaysia, 7 Centre for Phage Research, Department of Genetics and Genome Biology, University of Leicester, Leicester, United Kingdom

* F.Nobrega@soton.ac.uk

**Data Availability Statement:** All original code has been deposited at GitHub https://simrankushwaha.

## Abstract

Understanding the dynamic evolution of *Salmonella* is vital for effective bacterial infection management. This study explores the role of the flexible genome, organised in regions of genomic plasticity (RGP), in shaping the pathogenicity of *Salmonella* lineages. Through comprehensive genomic analysis of 12,244 *Salmonella* spp. genomes covering 2 species, 6 subspecies, and 46 serovars, we uncover distinct integration patterns of pathogenicity-related gene clusters into RGP, challenging traditional views of gene distribution. These RGP exhibit distinct preferences for specific genomic spots, and the presence or absence of such spots across *Salmonella* lineages profoundly shapes strain pathogenicity. RGP preferences are guided by conserved flanking genes surrounding integration spots, implicating their involvement in regulatory networks and functional synergies with integrated gene clusters. Additionally, we emphasise the multifaceted contributions of plasmids and prophages to the pathogenicity of diverse *Salmonella* lineages. Overall, this study provides a comprehensive blueprint of the pathogenicity potential of *Salmonella*. This unique insight identifies genomic spots in nonpathogenic lineages that hold the potential for harbouring pathogenicity genes, providing a foundation for predicting future adaptations and developing targeted strategies against emerging human pathogenic strains.

## Introduction

The interplay between conserved and variable features in bacterial genomes plays a crucial role in shaping the diversity and adaptability of different species [1]. Within a species, the core genome, comprising genes universally present, handles essential cellular functions. In contrast,

github.io/Genome-Plasticity-in-Salmonella/ and Zenodo, https://doi.org/10.5281/zenodo.12667378. An interactive visualisation of the gene content of the spots is also available at GitHub. The metadata of the isolates, phylogenetic analysis and the country of isolation can be visualised on Microreact, https://microreact.org/project/pRbGPKfTYKTHJfDipWbZze-project1-genomic-plasticity-is-a-blueprint-of-diversity-in-salmonella-lineages and https://microreact.org/project/rxxw1HJL7CGqSRqifU6q3W-project2-genomic-plasticity-is-a-blueprint-of-diversity-in-salmonella-lineages.

**Funding:** This work was supported by British Council Newton Bhabha Fund [grant number 654669088] to S.K.K and Wessex Medical Trust [grant number AB03] to F.L.N. The funders had no role in study design, data collection and analysis, decision to publish, or preparation of the manuscript.

**Competing interests:** The authors have declared that no competing interests exist.

**Abbreviations:** ABR, antibiotic resistance; HTH, helix-turn-helix; IHF, integration host factor; LPS, lipopolysaccharide; MGE, mobile genetic element; PATRIC, PathoSystems Resource Integration Center; pef, plasmid encoded fimbriae; QAC, quaternary ammonium compound; RGP, regions of genomic plasticity; RM, restriction-modification; spv, *Salmonella* plasmid virulence; T3SS, type III secretion system.

the flexible genome consists of genes that vary between individual strains, allowing bacteria to adapt to specific environments and acquire pathogenic traits [2–4]. These variable genes are often organised into regions of genomic plasticity (RGP) [5], which are regions of a genome structurally absent in other related genomes and typically associated with frequent rearrangements, such as those mediated by mobile genetic elements (MGEs). These elements serve as potent facilitators for acquiring genes related to virulence, antibiotic and stress resistance, and anti-phage immunity, contributing to the dynamic evolution of the bacterial genome [6–9]. Exploring this genomic plasticity is crucial for understanding bacterial evolution, phylogeny, and pathogenic potential.

*Salmonella* offers an excellent model for studying these variable genomic features. Its diverse spectrum of species, subspecies, and serovars showcases the inherent flexibility in its genome, a pivotal factor in shaping both the phylogeny and pathogenic potential of *Salmonella* [10–12]. Consequently, exploring the genomic plasticity of *Salmonella* becomes a key avenue for gaining insights into its evolution as a pathogen.

To gain further understanding of the structural and functional features of RGP in *Salmonella*, we carried out a comprehensive analysis of 12,244 *Salmonella* spp. genomes. Our findings revealed that gene clusters associated with virulence, stress resistance, antibiotic resistance, and anti-phage defence exhibit specific preferences for RGP integrated into distinct genomic spots. These preferences seem to be influenced by neighbouring genes that likely share regulatory and functional coordination. The irregular distribution of these genomic spots across diverse *Salmonella* lineages establishes a blueprint for pathogenicity and survival strategies. Deciphering the complex interplay between pathogenicity-related gene clusters and RGP not only improves our understanding of *Salmonella* evolution, but also enables us to uncover novel pathogenicity genes, anticipate future adaptations, and identify targets for disease prevention, management, and therapeutic interventions.

## Results

### The mobilome of *Salmonella* is highly variable across lineages

MGEs play a pivotal role in driving genetic diversity and shaping the evolutionary trajectories of bacteria, enabling them to adapt to various environmental challenges [13]. One significant way through which MGEs exert their influence is by facilitating the horizontal transfer of genes associated with pathogenicity traits, directly impacting the potential of bacterial pathogens. To determine the broad relevance of specific MGEs in defining specific pathogenicity attributes of *Salmonella*, we explored the variation in plasmids and prophages across 12,244 *Salmonella* genomes (**S1 Table**). Our dataset included representative strains from the 2 species and 6 subspecies of *Salmonella*, as well as 46 serovars of *Salmonella enterica* subsp. *enterica* (**Fig 1A**). As expected, the most prevalent serovars were Typhi (2,440 strains) and Typhimurium (2,170 strains), the main causative agents of typhoid fever in humans [14] and typhoid fever/gastroenteritis [15] in various animals and humans, respectively. The genome sequence of these strains was used to infer a phylogenetic topology representing the genomic diversity within the genus *Salmonella* (**Fig 1B**). The overall topology of the phylogeny is in accordance with the phenogram created previously from concatenated MLST genes of a smaller number of genomes [16].

Analysis of plasmid prevalence across different *Salmonella* subspecies and serovars revealed a variable abundance of the plasmid contigs (i.e., contigs that represents a part or the entirety of a plasmid) (**Fig 1C and S2 Table**). For example, serovar Kentucky, Gallinarium, and Oranienburg exhibited an average of 15 plasmid contigs, whereas most strains of serovar Choleraesuis had an average of 2 plasmid contigs. Among the identified plasmids, the most prevalent

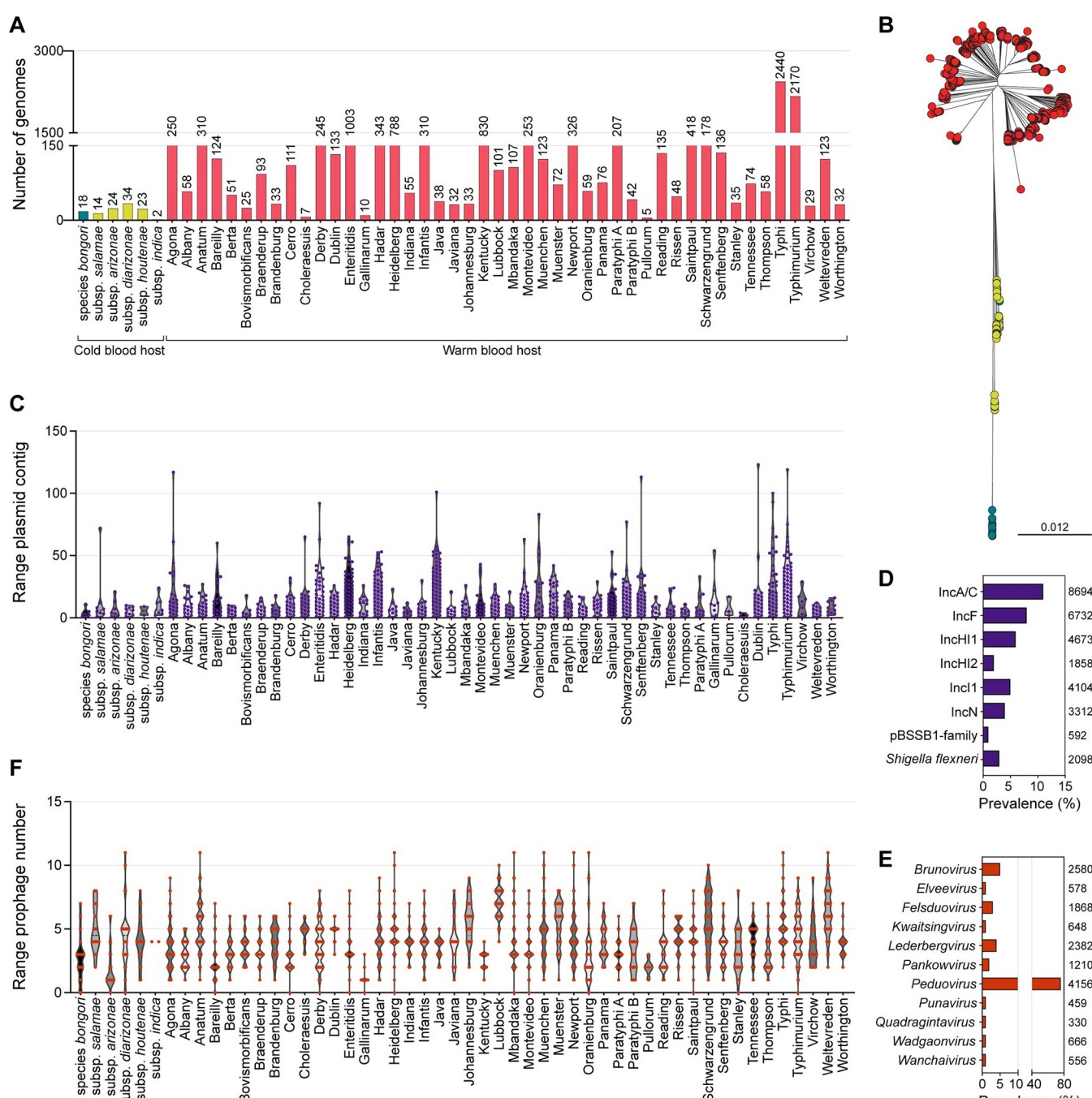

**Fig 1. Characterisation of the *Salmonella* database. (A)** Distribution of strains from various *Salmonella* subspecies and serovars. The number of genomes for each specific *Salmonella* subspecies and serovar is shown on top of the bars. *Salmonella* subspecies infecting cold-blooded hosts are highlighted in shades of green and *S. enterica* subsp. *enterica* serovars infecting warm-blooded hosts are indicated in red. **(B)** Phylogenetic distribution of the *Salmonella* strains with a colour scheme analogous to (A). The scale bar represents a phylogenetic distance of 0.012 nucleotide substitutions per site. **(C)** Range of plasmid contig number in *Salmonella* subspecies and serovars. **(D)** Prevalence of plasmid incompatibility groups in *Salmonella*. The *Shigella flexneri* plasmid incompatibility group refers to virulence plasmid pINV. The number of plasmid contigs for each plasmid incompatibility group is indicated to the right of the bars. **(E)** Prevalence of prophage genera in *Salmonella*. The number of prophages for each genus is indicated to the right of the bars. **(F)** Range of prophage content in *Salmonella* subspecies and serovars. The data underlying this figure can be found in S1, S2, and S5 Tables and S1 File.

were those belonging to the IncA/C group (11%, 8,694), and IncF group (8%, 6,732) (**Fig 1D** and **S3 Table**). Notably, IncA/C plasmids were predominantly found (35%, 3,056) in serovar Typhimurium. IncF plasmids were mostly identified in serovar Kentucky (28%, 1,866), while IncHI1 plasmids (32%, 1,481) were more commonly observed in serovar Typhi (**S3 Table**). Other plasmid types exhibited a more even distribution across different species. Analysis of prophage prevalence in *Salmonella* shows that the vast majority of strains (99.7%, 12,213) carry at least 1 prophage, accounting for a total of 52,134 prophage regions (**S4 Table**). From this total, the taxonomy of 2,928 complete dsDNA prophage regions could be determined using taxmyPHAGE (https://github.com/amillard/tax_myPHAGE). In most cases, we identified prophage regions associated with phages from multiple families, genera, and species, resulting in a total of 31,490 entries. All these phages belong to the kingdom *Heunggongvirae*, phylum *Uroviricota*, and class *Caudoviricetes*. Within *Caudoviricetes*, 69% (21,882) of the phage regions belong to the genus *Peduovirus*, 5% (1,668) to the genus *Lederbergvirus*, 5% (1,492) *Felsduovirus*, and 5% (1,454) to the genus *Brunovirus* (**Fig 1E**). The remaining phages are distributed across 43 other identified genera, though in smaller quantities (**S5 Table**). The most commonly identified phages show high similarity to *Salmonella* phage SEN34 (**S5 Table**). Similar to plasmids, the average number of prophages per strain varies among serovars, with serovar Lubbock averaging 7 prophages, whereas serovar Gallinarum has only 1 (**Fig 1F**).

In summary, our findings highlight the remarkable diversity observed in the mobilome of *Salmonella*. This diversity is reflected in the abundance and types of MGEs present per species, subspecies, and serovars. The variable nature of the mobilome and the resulting diversity in gene composition are expected to play a critical role in shaping the pathogenicity, adaptation, and distribution of *Salmonella*.

## Virulence determinants are more prevalent in chromosomal regions

We next analysed the presence of factors contributing to the survival and adaptation of *Salmonella* to the environment. These included a set of virulence factors, antibiotic resistance genes, stress response genes, and phage-resistance genes (i.e., anti-phage defence systems) (the complete list of genes can be found in **S6 Table**). This analysis revealed the presence of virulence factors predominantly in *S. enterica* subsp. *enterica*, with an average of 46 virulence factors per strain (**Fig 2A** and **S7 and S8 Tables**). *S. bongori* has the lowest number of virulence genes, 20. In comparison to most *S. enterica* subsp. *enterica* strains, *S. bongori* lacks the *Salmonella* pathogenicity island 2 (SPI-2), which encodes a type III secretion system (T3SS) that plays a central role in systemic infections and the intracellular phenotype of *S. enterica*, except for one strain (PATRIC genome ID 1173775.3) that also groups with cold-blooded subspecies of *S. enterica* in the phylogenetic tree (**Fig 1B**). The ability of SPI-2 to transfer to, and be functional in, *S. bongori* has been previously demonstrated experimentally [17] but, to our knowledge, not yet found in nature [16]. *S. bongori* do contain the SPI-1 with a T3SS that promotes invasion of epithelial cells through the secretion of different effector proteins [18]. SPI-1 is prevalent across all *Salmonella* species, subspecies, and serovars, but with variations in the presence of secreted effectors encoded by *spt* and *slr*, as well as *avr* and *ssp* genes, especially the latter (**S1 Fig**).

Serovars Gallinarum and Pullorum share a more recent common ancestor with the serovar Enteritidis, but they harbour a different virulence plasmid [19]. This difference is evidenced by the absence of gene *rck*, which is responsible for evading the host immune response and surviving inside the host [20], in both Gallinarum and Pullorum virulence plasmid (**S1 Fig**). Our analysis revealed that the majority of genes encoding virulence factors (97%) are located in chromosomal regions, rather than being carried by prophages or plasmids (**Figs 2B and S1**

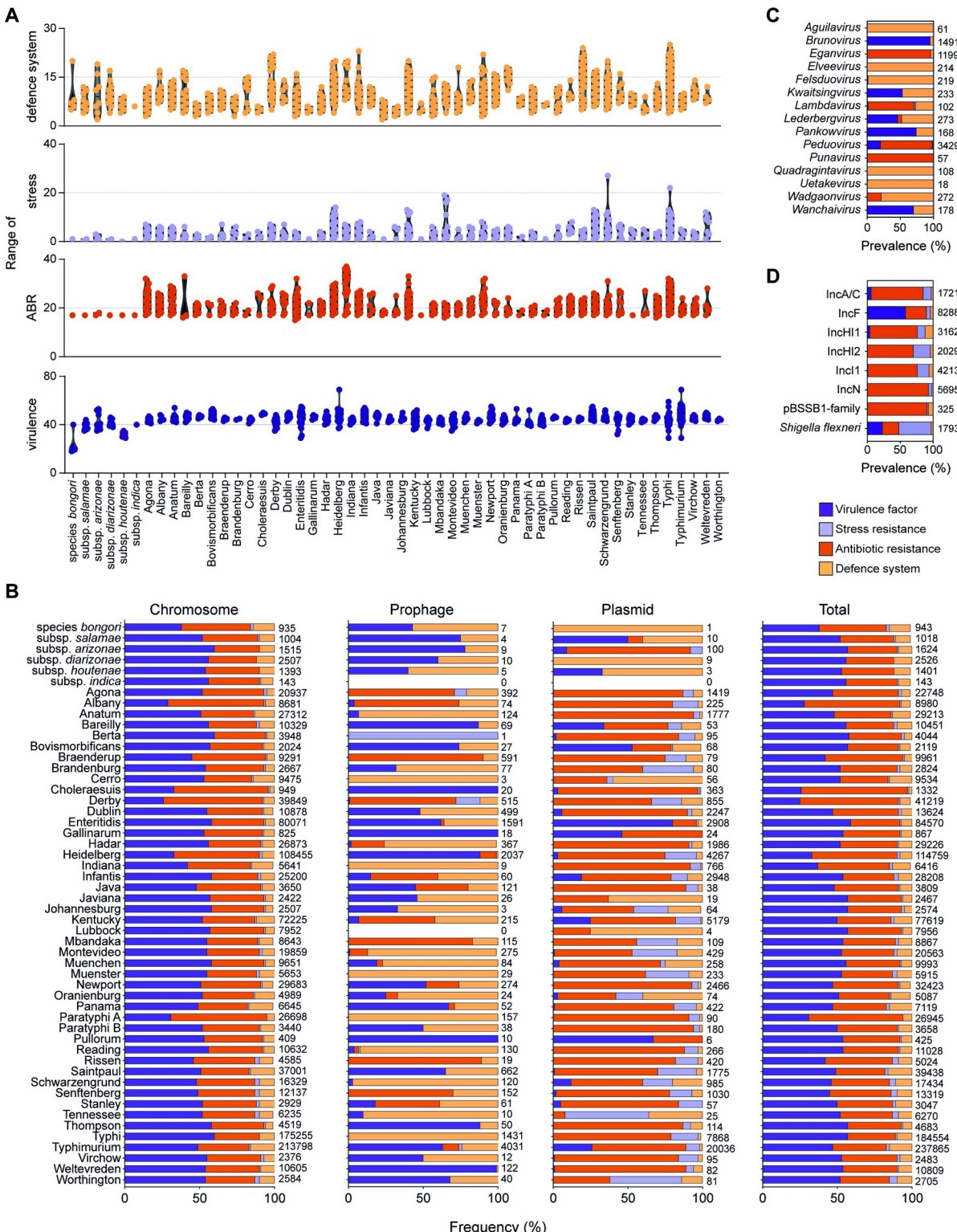

**Fig 2. Prevalence and distribution of pathogenicity determinants in *Salmonella*.** (**A**) Range of virulence factor, stress resistance gene, antibiotic resistance gene, and anti-phage defence system number across *Salmonella* subspecies and serovars. (**B**) Distribution of the pathogenicity determinants across chromosome, prophages, and plasmids. The total count for each category is shown at the right side of the bars. (**C**) Distribution of the pathogenicity determinants across prophage genera present in *Salmonella* at >1% abundance. (**D**) Distribution of the pathogenicity determinants across plasmid incompatibility groups present in *Salmonella* at >1% abundance. In all panels, virulence

factors, stress resistance genes, antibiotic resistance genes, and defence systems are coloured according to the key. The data underlying this figure can be found in S7, S8, and S9 Tables.

and **S7 Table**). However, certain virulence genes are more commonly associated with prophages or plasmids. For example, genes *sod* and *grv*, critical to the bacterial response to oxidative stress and their ability to survive within immune cells [21,22], are frequently found in prophage regions, mostly of the *Peduovirus* (>80%) genus (**S1 Fig**). Among the prophages identified, only *Brunovirus* seem to preferentially carry virulence factors (**Fig 2C**). Surprisingly, in contrast to existing literature [23], we found that the majority of *gog* genes, which are associated with an anti-inflammatory function [24], are located on the chromosome (1,574 out of 1,703) rather than a prophage region (**S1 Fig and S7 Table**). The well-known virulence genes *spv* (involved in intracellular survival and evasion of the host immune response [25]), *pef* (plasmid encoded fimbriae, important for colonisation of the host and establishment of infection [26,27]), and *rck* (contributing to evasion of the host immune response [28]) are predominantly (97%, 99%, and 99%, respectively) found on plasmids (**S1 Fig**), particularly those belonging to the IncF group (**Fig 2D and S9 Table**). These *Salmonella* virulence plasmids (pSV) containing the *spv* genes were identified in *S. enterica* subsp. *arizonae* and *S. enterica* subp. *enterica* serovar Typhimurium, Dublin, Enteritidis, Choleraesuis, Gallinarum, and Pullorum, consistent with the existing literature [29–32]. Genes *pef* and *rck* have also been reported previously in pSV [33]. The *fyu* and *ybt* genes involved in iron acquisition [34] were predominantly associated with IncA/C plasmids. Notably, we did not find any virulence factors on IncHI2, IncN, and pBSSB1-family plasmids (**Fig 2D** and **S9 Table**).

In summary, our analysis reveals that virulence factors in *Salmonella* are primarily found within chromosomal regions, but specific gene clusters are preferentially located in prophages of the *Brunovirus* and *Peduovirus* genera, and plasmids of the IncF group.

## Antibiotic resistance determinants are primarily located in the chromosome

Antibiotic resistance (ABR) determinants were predominantly found in the chromosome (84%) (**S7** and **S10 Tables**). This high prevalence is primarily driven by widespread resistance to aminocoumarin, aminoglycoside, carbapenem, cephalosporin, cephamycin, fluoroquinolone, glycylcycline, macrolide, monobactam, nitroimidazole, penam, penem, peptide, phenicol, rifamycin, tetracycline, and triclosan antibiotics across all *Salmonella* subspecies and serovars (**S1 Fig**). These resistances are largely attributed to mutations in conserved genes, such as *cpr*, *cpxA*, and *hns*, which regulate multidrug efflux pump expression [35–38] (**S10 Table**).

Additionally, 16% of ABR determinants were located in plasmid contigs. These plasmids serve as the primary reservoir for resistance against β-lactam, diaminopyrimidine, gentamycin, kanamycin, streptomycin, sulphonamide, and trimethoprim antibiotics (**S1 Fig**). Among the plasmid schemes, all carried ABR, with IncN and the pBSSB1-family being most frequently associated with these determinants (**Fig 2D** and **S9 Table**). Less than 1% of ABR determinants were associated with prophages, mostly from the *Eganvirus*, *Punavirus*, *Peduovirus*, and *Lambdavirus* genera (**Fig 2C**).

Importantly, resistance to colistin, an antibiotic of last resort, was detected in 2.4% (288) of strains belonging to *S. enterica* subsp. *enterica*, with a predominant occurrence in serovars Saintpaul, Cholerasuis, and Paratyphi B (**S1 Fig**).

In summary, our results indicate that chromosomal mutations in various *Salmonella* genes confer ABR, reinforce the role of plasmids in influencing ABR patterns, highlighting plasmids of all schemes are drivers of ABR dissemination in *Salmonella*.

## Stress resistance genes are primarily located on plasmids and chromosomal regions

The presence of acid, biocide, and heavy metal resistance genes is closely linked to the maintenance and spread of antimicrobial resistance [39–41]. Interestingly, we observed that 2 multidrug-resistant serovars, Indiana and Rissen, exhibit the highest prevalence of *qac* genes, which are small multidrug resistance efflux proteins associated with increased tolerance to quaternary ammonium compounds (QACs) and other cationic biocides [42] (**S1 Fig**). *qac* genes are generally found in MGEs, particularly plasmids; here, they were found on IncA/C plasmids in over 25% of the cases (**S9 Table**).

Curiously, most strains in our dataset do not carry any heat-resistant genes (*hde*, *hsp*, *kef*, *psi*, *shs*, *trx*, and *yfd*), except for a small percentage (<20%) of strains from serovars Montevideo, Senftenberg, and Worthington, and the majority of these genes are located on IncA/C plasmids. On the other hand, *Salmonella* strains commonly exhibited resistance to heavy metals, with approximately 80% of the strains carrying genes conferring resistance to gold (**S1 Fig**). The only exceptions are *S. enterica* subsp. *houtenae* and *S. enterica* subsp. *enterica* serovars Typhi and Paratyphi A, which do not carry the *gol* cluster responsible for gold resistance. Serovars Heidelberg and Infantis show a high incidence (>95%) of arsenic resistance genes (*ars*), while serovars Tennessee, Rissen, Schwarzengrund, Worthington, and Senftenberg exhibit frequent (>80%) copper (*pco*) and silver (*sil*) resistance (**S1 Fig** and **S7 Table**).

Genes that confer resistance to heavy metals such as gold and arsenic are predominantly located in chromosomal regions, while those associated with mercury and tellurite resistance are commonly found on plasmids (IncA/C and *Shigella flexneri* [43] plasmid schemes for mercury, and IncHI1 and IncHI2 for tellurite) (**S1 Fig** and **S9 Table**). Among the different plasmid schemes, *Shigella flexneri* (pINV virulence plasmids) and IncHI2 plasmids are those most frequently associated with stress resistance genes, but IncA/C plasmids, due to their abundance, are responsible for the movement of most stress resistance genes (**Fig 2D** and **S9 Table**).

In summary, our findings reveal that genes associated with resistance to heat and heavy metals such as mercury and tellurite are primarily found on plasmids, while resistance to gold and arsenic is commonly found within chromosomal regions.

## Anti-phage defence systems are more prevalent in chromosomal regions

Anti-phage defence systems were found to be prevalent among *Salmonella* strains, with an average of 8 defence systems per strain. This is higher than the average found in *Escherichia coli* (6) [44] or *Pseudomonas aeruginosa* (7) [45] in previous studies. However, there is considerable variation in the number of defence systems carried by different subspecies and serovars. For example, serovars Typhimurium [17], Saintpaul [15], Panama [15], and Indiana [14] exhibit the highest prevalence of defence systems, while serovars Berta, Javiana, and Johannesburg have the lowest [4] (**Fig 2A** and **S7 and S8 Tables**).

Among the 90 defence system subtypes identified in *Salmonella* strains, the most prevalent were the restriction-modification (RM) and type I-E CRISPR-Cas systems, which are present in almost all subspecies and serovars (**S1 Fig**). However, the CRISPR-Cas system is absent in serovars Brandenburg, Lubbock, and Worthington. We noted significant variation in the prevalence of other defence systems across the *Salmonella* genus (**S1 Fig**). Each serovar appears to have a distinct profile of defence systems, suggesting selection of the most beneficial systems in specific environments or host interactions, as previously observed for distinct *E. coli* phylogroups [44]. For example, in serovars Typhi and Paratyphi A, the 3HP and Druantia type III systems are highly abundant. On the other hand, in Typhimurium, we observed a predominance of the BREX type I, Mokosh type II, PARIS types I and II, and Retron II-A defence

systems. Strains of Enteritidis exhibit an enrichment in CBASS type I, while Gallinarium and Pullorum frequently harbour Mokosh type I in addition to CBASS type I. Additionally, we found specific defence systems enriched in particular species and subspecies. For example, dCTP deaminase is more prevalent in *S. bongori*, Septu type I in *S. enterica* subsp. *indica* and *salamae*, and Gabija in *S. enterica* subsp. *arizonae* (**S1 Fig**).

In general, defence systems, including the abundant RM and CRISPR-Cas systems, are more frequently found within chromosomal regions (94%, **Fig 2B**) compared to prophages or plasmids. However, prophages of the *Aguilavirus*, *Elveevirus*, *Felsduovirus*, *Quadragintavirus*, *Uetakevirus*, and *Wadgaonvirus* show a clear preference for carrying defence systems over other types of pathogenicity-related genes (**Fig 2C**), and the defence systems 3HP, AbiL, BstA, Kiwa, Retron types I-A, I-C, and VI are predominantly found on prophages (**S1 and S2A Figs**). Other defence systems, such as AbiQ, Bunzi, Gao_19, Lit, PifA, ppl, retron type V, SoFic, and tmn are frequently associated with plasmids. When located on plasmids, the system Gao_19 (34%) is primarily linked to IncA/C plasmids, while PifA is mostly found (34%) on IncI1 plasmids. Lit (33%), Bunzi (50%), and tmn (37%) are often identified on IncHI1 plasmids, and ppl (30%) and SoFic (31%) are mostly associated with IncF plasmids (**S2B Fig and S9 Table**). Interestingly, although plasmids of all types often accommodate a greater abundance of other pathogenicity-related elements (**Fig 2D**), it is noteworthy that the IncHI1 plasmids demonstrate a higher inclination toward carrying defence systems compared to other plasmid schemes.

In summary, anti-phage defence systems are widespread in *Salmonella*, with a notable prevalence of the R-M and the CRISPR-Cas systems. The significant variation in defence system repertoire across *Salmonella* species and serovars observed here highlights the significance of these systems in the evolution and adaptation of this pathogenic bacterium.

## Gene clusters integrate into preferential spots in the *Salmonella* genome

Our analysis uncovered substantial variability in the presence and arrangement of genes associated with virulence, antibiotic resistance, stress response, and anti-phage defence genes among different *Salmonella* strains. This variability strongly suggests the occurrence of genomic rearrangements involving the insertion and deletion of genes. To gain a deeper understanding of genome plasticity within *Salmonella*, we performed a comprehensive mapping analysis using PPanGGoLiN [46] and panRGP [47] to identify RGP.

The pangenome analysis categorises gene families as core (conserved in all genomes), persistent (found in more than 90% of genomes), shell (present in 50% to 90% of genomes), and cloud (found in less than 50% of genomes). Our findings show that only 4.6% of the gene families are conserved in over 90% of *Salmonella* genomes (3,575 persistent gene families, including 65 core families), while 5.5% of the gene families (4,256) were present at intermediate frequencies (shell), and approximately 90% (69,678) at low frequency (cloud) (**Fig 3A**).

Analysis of gene length shows that core gene families range from 195 to 3,699 bp, persistent gene families from 93 to 17,610 bp, shell gene families from 93 to 10,137 bp, and cloud gene families from 93 to 24,540 bp (**Fig 3B**). The median gene length showed that persistent (873 bp) and core genes (729 bp) are significantly longer than shell genes (558 bp) and cloud genes (420 bp) (**Fig 3B and S11 Table**). In eukaryotes, longer genes are suggested to be more evolutionarily conserved and associated with important biological processes [48–51]. This observation aligns with our findings in *Salmonella*, as functional analysis of the persistent genes revealed their essential roles in survival and fitness (**S11 Table**). In contrast, shorter gene length is associated with high expression [52], providing an advantage in response to stimuli [53]. This observation is consistent with the role of accessory shell and cloud genes, which are

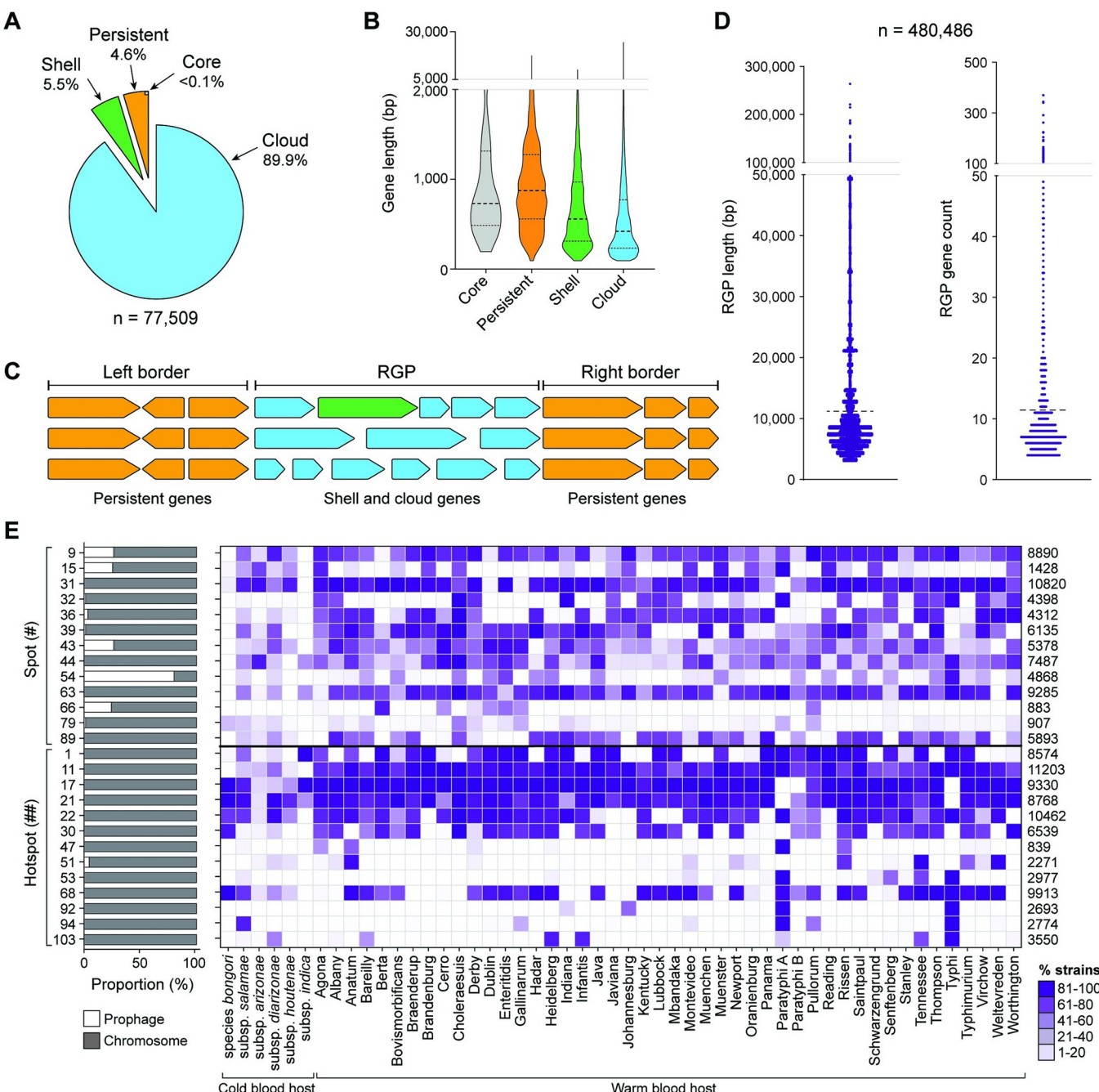

**Fig 3. Pangenome analysis of 12,244 *Salmonella* genomes. (A)** Partition of *Salmonella* gene families by PPanGGOLiN, based on their conservation across strains. Core, conserved in all genomes (grey); persistent, conserved in almost all genomes (orange); shell, moderately conserved (green); cloud, poorly conserved (blue). **(B)** Length of core, persistent, shell, and cloud gene families. Thicker dotted lines represent the median length, while thinner dotted lines indicate the upper and lower quartiles. **(C)** Schematic representation of RGP, consisting of variable shell (green) and cloud (blue) genes, identified in between conserved border genes (persistent, orange). **(D)** Length and gene count in the identified RGP. Each dot indicates an individual RGP length and RGP gene count, respectively. Dotted lines indicate the average for each category. "*n*" indicates the total number of RGP represented. **(E)** Heatmap of the distribution of 26 integration spots across chromosome (grey) and prophages (white), and their prevalence across *Salmonella* subspecies and serovars. The 26 spots correspond to those identified in >1% strains of *Salmonella* containing pathogenicity determinants. Spots (#) are characterised by the presence of >1 and ≤100 unique RGP families with diverse gene content, and hotspots (##) by the presence of >100 unique RGP families. The data underlying this figure can be found in S11, S12, and S13 Tables.

likely to confer fitness benefits under specific environmental and stress conditions. Clusters of the less conserved shell and cloud gene families form RGP, which primarily result from horizontal gene transfer events. To study the dynamics of RGP, these regions from different genomes can be grouped into specific insertion spots based on the presence of conserved flanking persistent genes (**Fig 3C**). Our analysis identified a total of 673,113 RGP, among which 71.4% (480,486) were clustered in 1,345 spots. The RGP not assigned to a specific spot on the chromosome, either due to missing border genes or to being plasmid contigs, were excluded from further analysis as they do not meet the criteria for true RGP. The RGP associated with chromosomal spots have an average length of 11,182 bp (range: 3,001 to 355,687 bp) and an average of 11 genes (range: 4 to 522) per RGP (**Fig 3D**). The majority of the RGP (96.5%) is located in the bacterial chromosome and the remaining 3.5% are prophages (i.e., border genes of the spot correspond to those of the prophage) (**S12 Table**). Out of the 1,345 spots, 74.65% (1,004) were specific to a single type of RGP, while the remaining spots exhibited the potential to harbour a diverse array of RGP families with diverse gene content. Importantly, 1.64% [22] of these spots could harbour >100 distinct RGP families (**S13 Table**), suggesting higher rates of gene acquisition and underscoring these regions as hotspots for gene integration [47].

We screened all spots for the presence of virulence genes, antibiotic resistance genes, stress resistance genes, and defence systems (**S14 Table**). Among the resulting 266 spots, we selected those with variable content present in at least 1% of the strains, yielding 26 spots (#) (13 of which are hotspots, ##) for further analysis. Some spots were relatively specific to certain serovars, such as hotspot ##47 in serovars Paratyphi A, Anatum, Agona, and Rissen; hotspot ##92 in Typhi, Paratyphi A, and Johannesburg; or hotspot ##94 in Typhi and Paratyphi. In contrast, other spots were widely distributed, such as spot #9 or hotspot ##22 (**Fig 3E**).

When examining the gene content of these spots related to virulence, stress, antibiotic resistance, and defence systems, we can observe that genes with specific functions show a clear propensity to congregate within particular locations (**Fig 4A**). For example, virulence genes tend to favour hotspots ##21, ##30, ##92, and ##94, while defence systems seem to prefer hotspots ##22 and ##68. However, the specific gene content of these spots can vary, as exemplified in **Fig 4B** and detailed in **Fig 4C**. Specific gene families also show preferences for particular spots. For example, *lfp* genes preferentially localise in hotspot ##30, while *fae* genes predominantly localise in spot #36 (**Fig 4C** and **S14 Table**). The gene cluster conferring tolerance to gold (*gol*) distinctly favours hotspot ##17; the absence of this spot in serovars Typhi and Paratyphi A leads to the absence of gold resistance (**S1 Fig**). However, hotspot ##17 is present in a few strains of *S. enterica* subsp. *houtenae*, where gold resistance is lacking, indicating that the presence of this spot does not consistently correlate with gold resistance. Similarly, the gene cluster conferring arsenic resistance (*ars*) predominantly localises in hotspot ##103, which is prevalent in serovars Heidelberg and Infantis, the strains of which display the highest prevalence of arsenic resistance. However, hotspot ##103 is also frequently found in serovar Typhi, where arsenic resistance is absent. Collectively, these findings underscore that the presence of a spot where a specific gene cluster predominantly localises does not unequivocally signify the presence of said gene cluster; conversely, the absence of the site often corresponds to the absence of the specific gene cluster. In cases of the former, other influencing factors may contribute to the selection for the presence of such gene clusters, potentially spurred by environmental pressures.

Another important example of gene cluster preference for distinct spots involves the type I-E CRISPR-Cas system, predominantly found in hotspot ##22. Hotspot ##22 is ubiquitously present across all species except for serovars Brandenburg, Java, Javiana, Johannesburg, Lubbock, Mbandaka, Panama, Reading, and Worthington (**S14 Table**). In these serovars, the

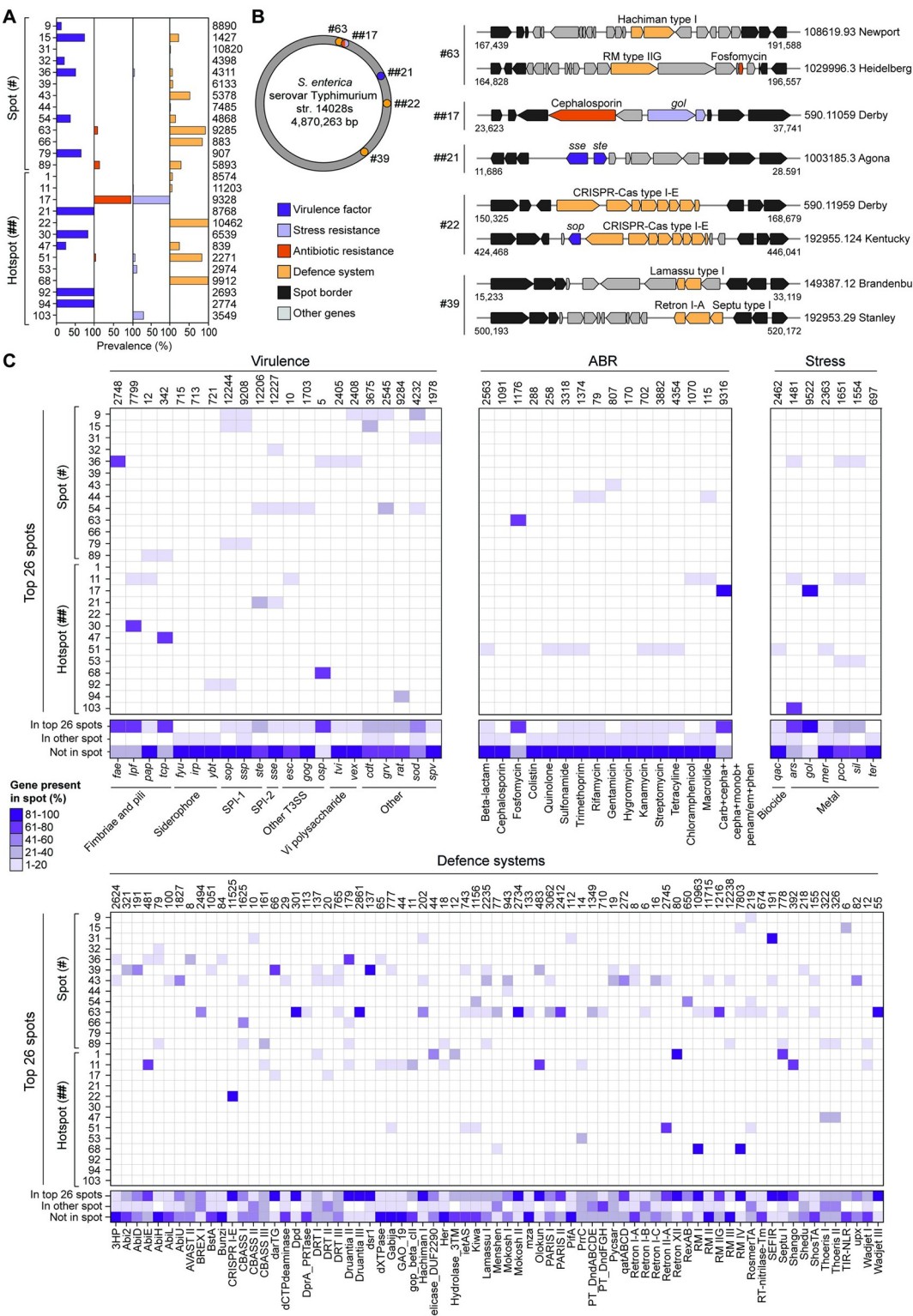

**Fig 4. Distribution of gene content in the spots of *Salmonella* genomes. (A)** Frequency of pathogenicity determinants across spots identified in at least 1% strains of *Salmonella*. The number of strains containing each spot is indicated at the right of the bars. (**B**) Mapping of spots present on the reference strain *S. enterica* subsp. *enterica* serovar Typhimurium str. 14028s, with examples on the right illustrating various gene arrangements. The accession number, serovar, and genomic location are provided. In panels (A) and (B), virulence factors, stress resistance genes, antibiotic resistance genes, defence systems, other

RGP genes, and spot border genes are colour-coded according to the key. The category of other RGP genes represents those genes within the RGP that are not the specific pathogenic determinants under study. **(C)** Heatmap of the frequency at which a specific gene cluster appears outside a spot, in one of the 26 most abundant spots, or in other spots. Only gene clusters found in the 26 most abundant spots are depicted. The total count of gene cluster instances found across the *Salmonella* genus is shown on top. The data underlying this figure can be found in S14 Table. An interactive visualisation of the gene content in the spots can be found in https://doi.org/10.5281/zenodo.12667378.

CRISPR-Cas system is either entirely absent or present in only a limited number of strains (**S1 Fig**). The well-conserved nature of the type I-E CRISPR-Cas system in *Salmonella* [54,55] seems intrinsically tied to the widespread prevalence of hotspot ##22 (90% of all strains). Additionally, RM types I and IV have a strong preference for hotspot ##68, which is present in over 80% of the strains, thus accounting for the wide prevalence of RM systems in *Salmonella*.

On a broader scale, we also observe an inclination for pathogenic determinants with related functions to cluster within the same spots (13 out of 26). This is well demonstrated by spots housing diverse anti-phage defences (e.g., ##11, #39, #43, and #63). For example, spot #63 contains an array of defence systems, with Dpd, Druantia type III, Mokosh type II, and Wadjet type III present in more than 80% of instances (**Fig 4A and 4C** and **S14 Table**).

In our study, we also observed that the majority of gene clusters are not associated with specific genomic spots (**Fig 4C**). This lack of association can be attributed to several factors. Firstly, many gene clusters, including some responsible for stress resistance (e.g., *qac*, *mer*, *ter*) and antibiotic resistance (e.g., β-lactam, chloramphenicol, streptomycin, among others), are predominantly located on plasmids rather than chromosomal spots (**S1 Fig**). Plasmids often serve as vectors for horizontal gene transfer, contributing to their widespread distribution across different strains without a fixed genomic position. Additionally, virulence genes such as *spv*, *fyu*, *irp*, and *ybt*, along with defence systems like PifA, Gao_19, and Bunzi also frequently reside on plasmids, further explaining their absence from chromosomal spots (**S1 Fig**). Conversely, certain gene clusters, such as *sop* and *ssp* from SPI-1 [18] and defence systems RMII and RMIII (**S1 Fig**) are ubiquitous in *Salmonella* genomes. Their consistent presence across various strains likely precludes their identification in variable spots in our analysis.

In summary, our findings reveal that certain pathogenic determinants display a preference for integration into specific spots within the *Salmonella* genome. The presence of these spots indicates the potential presence of the associated pathogenic determinants. These genomic spots may potentially play critical roles in bacterial adaptability and fitness, as evidenced by the exclusive association of the type I-E CRISPR-Cas system with serovars containing hotspot ##22, warranting future experimental investigation.

## Spot flanking genes are likely determinants of gene cluster preference

We investigated whether the preference of the pathogenicity gene clusters for particular spots was influenced by the gene context, particularly focusing on the highly conserved (persistent and core) genes flanking the spots. To accomplish this, we examined the genomic locations of the 26 prevalent spots identified in our study (an interactive visualisation of the gene content of the spots can be found at the associated Github, see Data Availability).

While some flanking genes have unknown functions, their predicted roles suggest potential connections to spot functionality. For example, hotspot ##1 and spot #63, which seem to be preferred by anti-phage defence systems, are associated with helix-turn-helix (HTH)-type transcriptional regulators (*ecpR* and *gntR*, respectively) as border genes (**S15 Table**). Gene *ecpR* in hotspot ##1 is negatively regulated by H-NS and positively regulated by itself and integration host factor (IHF), a protein involved in various phage-related processes such as integration and propagation [56]. The regulatory interactions involving *ecpR* and IHF suggest a

potential influence of the first on phage-related processes. Similarly, spot #63 features the flanking gene *gntR*, which influences cell wall permeability and bacterial motility, both factors known to affect phage infectivity [57]. Additionally, spot #63 contains a flanking gene encoding protein L-threonine 3-dehydrogenase, and L-threonine has been observed to impact phage infection in *E. coli* [58]. Hotspot ##22 houses the CRISPR-Cas type I-E defence system and is adjacent to the *cysD* gene involved in cysteine biosynthesis [59]. Notably, the regulation of the *cysD* gene involves the *cnpB* gene, which participates also in CRISPR-Cas regulation [60]. This co-localisation suggests potential coordination between these genes for regulatory purposes.

Hotspot ##17 is a hotspot for gold resistance and various antibiotic resistance genes and is flanked by the *oprM* gene. *oprM* encodes an outer membrane protein that functions as an antibiotic and metal pump [61], enabling the bacterium to defend against the toxicity of antibiotics and metals. This suggests that the presence of *oprM* in the vicinity of hotspot ##17 contributes to gold and antibiotic resistance by facilitating their efflux from the bacterial cell. Similarly, hotspot ##53, associated with copper and silver resistance genes, is adjacent to the *uspB* gene encoding a universal stress response protein [62]. UspB promotes cell survival and protects against stress-induced damage, potentially aiding the bacterium in coping with copper and silver stress.

Hotspot ##21 contains the *ste* and *see* genes involved in *Salmonella* enterotoxin production and is flanked by the *mdoD* gene, which encodes glucan biosynthesis protein D. This protein is essential for the synthesis of osmoregulated periplasmic glucans, which contribute to the stability and integrity of the bacterial cell envelope [63]. Although no direct connection between periplasmic glucans and enterotoxin production has been reported, it is possible that glucan production and the secretion systems responsible for enterotoxin export indirectly influence each other through broader cellular processes or regulatory networks. Hotspot ##30 contains the *lpf* gene cluster responsible for the production of long polar fimbriae, which facilitates bacterial adherence and colonisation of host cells and tissues [64,65]. This spot is flanked by the *eptB* gene encoding Kdo(2)-lipid A phosphoethanolamine 7"-transferase, an enzyme that modifies the lipopolysaccharide (LPS) lipid A portion, contributing to bacterial resistance against cationic antimicrobial peptides [66]. *eptB* may support the survival of *Salmonella* adhering to host cells via *lpf* by protecting the bacterial cells from the antimicrobial peptides produced by endothelial cells, particularly in environments like the gastrointestinal tract.

Finally, hotspot ##51, which harbours several antibiotic resistance genes, is flanked by the *yidC* and *mdtL* genes. The *yidC* gene encodes the membrane protein YidC, crucial for the insertion and folding of membrane proteins in bacteria. YidC is implicated in the proper folding and assembly of essential membrane proteins associated with antibiotic resistance mechanisms and has been proposed as a potential antibiotic target [67,68]. On the other hand, the *mdtL* gene encodes a multidrug efflux transporter protein responsible for exporting a wide range of drugs and toxic compounds out of the bacterial cell, contributing to antibiotic resistance.

In conclusion, our investigation revealed potential relationships between spot functionality and the genes in their vicinity. The identification of specific flanking genes suggests their involvement in various processes related to phage defence, metal resistance, stress response, and antibiotic resistance. These spatial arrangements provide insights into potential coordination, regulatory connections, and adaptive mechanisms within bacterial genomes.

## Discussion

The genetic landscape of *Salmonella* is a mosaic shaped by various factors, with RGP acting as significant contributors. These dynamic genomic segments house diverse gene clusters that

hold the potential to dictate the genetic makeup of *Salmonella* strains. In the traditional view, gene distribution within a genome was often perceived as a stochastic process, but recent insights have challenged this view by revealing that genes linked to specific functions tend to cluster in certain regions [69,70]. The extent and implications of this phenomenon for bacterial evolution and adaptation have remained largely unexplored.

Our findings revealed a distinctive pattern of nonrandom integration of gene clusters into specific RGP of *Salmonella*. Exploring the distribution of certain RGP across diverse lineages of *Salmonella* revealed their pivotal role in shaping genetic content, and thus the pathogenicity and survival strategies of each lineage. Noteworthy examples include the presence of SPI-2 in 1 strain of *S. bongori*, which may provide it with the ability to infect warm-blooded hosts [17]. This divergence challenges established notions of gene distribution and exemplifies how RGP can redefine our understanding of gene presence and absence across lineages. Furthermore, the association between the absence of type I-E CRISPR-Cas system and the lack of hotspot ##22 provides a rationale for the previous observation of the missing type I-E CRISPR-Cas system in specific *Salmonella* serovars [71].

The mobility of gene clusters across genomes has raised questions about their potential integration into sites lacking proper regulation of expression [72–75]. Potential issues arise when the cluster relies on regulatory interactions absent in the new host, leading to incorrect gene expression, or when auxiliary interactions and dependencies on the host become relevant [76]. However, our study demonstrates a nonrandom integration pattern of RGP and their associated gene clusters, suggesting a purposeful selection of locations rather than randomness. Bacterial genomes are often organised in gene clusters regulated by shared regulators [77], supporting the idea that RGP are placed in specific spots primarily due to the benefit of co-regulation. This suggests that genes flanking certain genomic spots might dictate the integration of particular RGP. For example, the strategic insertion of genes responsible for long polar fimbriae production in regions flanked by antimicrobial peptide resistance genes suggests functional synergy, potentially aiding survival of *Salmonella* by protecting it from host-produced antimicrobial peptides during invasion. This underscores the likely coordination of gene expression among co-localised genes. Similarly, the positioning of stress resistance genes near specific RGP, harbouring metal resistance genes (##17, #53) might reflect a fine-tuned regulatory network to efficiently counter stressors. Notably, hotspots associated with antibiotic resistance genes (e.g., ##51) are flanked by genes implicated in antibiotic resistance mechanisms, while certain spots with anti-phage defence systems (##1, #63) are flanked by HTH transcriptional regulators linked to phage-related processes. These preferences for genomic locations might be driven by selective pressures or other factors that ensure co-expression and coordinated functionality, contributing to the intricate landscape of bacterial adaptation and evolution. Examining these potential functional links can unveil novel pathogenicity traits and gene interaction networks crucial for understanding *Salmonella* pathogenicity.

Unsurprisingly, our findings underscored the prominent role of chromosomal mutations and plasmids in influencing ABR patterns within *Salmonella*. A substantial proportion (84%) of ABR determinants were housed in the chromosome and 16% within plasmids, particularly those of the IncA/C and IncN plasmid groups. Noteworthy variations were observed across subspecies and serovars, with ABR genes predominantly concentrated in *S. enterica* subsp. *enterica*, raising concerns over the potential amplification of ABR due to human antibiotic usage. An especially troubling discovery was the presence of colistin resistance genes within serovars associated with human outbreaks, such as Saintpaul, Cholerasuis, and Paratyphi B. Colistin, designated as a crucial antibiotic by the World Health Organization [78], serves as a last-resort defence against life-threatening infections caused by multidrug-resistant gram-negative bacteria [78]. The occurrence of plasmid-borne colistin resistance within these outbreak-

causing serovars [79] carries the risk of propagation to other bacteria, including those with substantial clinical relevance.

But the role of plasmids extends beyond ABR. The IncF group favours virulence factors, aligning with previous reports [80]. Plasmids from the IncA/C group are key vectors for *qac* gene dissemination, associated with antimicrobial and biocide resistance, and also carry metal resistance determinants. Plasmids affiliated with the *Shigella flexneri* preferentially carry stress resistance determinants (*qac*, *mer*, and *ter*), while those from the IncHI1 group emerge as prominent bearers of defence systems. The differential prevalence of these traits in *Salmonella* can be attributed to the distinct plasmid types prevalent in each lineage. For example, IncA/C are mostly found in *S.* Typhimurium and confer resistance against most pathogenic determinants, while IncN, found primarily in *S.* Typhi, exhibit resistance against beta-lactams [81].

Our study also reveals the involvement of prophages in contributing to pathogenicity-associated gene patterns within *Salmonella*, especially in the case of anti-phage defence systems. Moreover, specific phage genera are linked to the dissemination of other factors, with *Brunovirus*, *Pankowvirus*, and *Wanchaivirus* frequently carrying virulence genes; *Eganvirus*, *Lambdavirus*, *Peduovirus*, and *Punavirus* favouring ABR genes; and *Aguilavirus*, *Elveevirus*, *Felsduovirus*, *Quadragintavirus*, *Uetakevirus*, and *Wadgaonvirus* preferentially carrying defence system genes. Overall, our findings underscore the multifaceted contributions of plasmids and prophages in shaping the pathogenicity of diverse *Salmonella* lineages.

In the broader context, these findings offer a novel perspective on deciphering the evolutionary trajectory of *Salmonella*. Further studies focusing on the relationships between gene clusters, RGP, and pathogenic attributes could improve our understanding of the mechanisms driving the emergence of diverse *Salmonella* lineages. Such knowledge might help in predicting future adaptations and developing targeted interventions to combat infections. For example, the identification of genomic spots within strains of nonpathogenic lineages with the capacity to house RGP encoding human pathogenicity genes could help us to foresee evolutionary shifts and formulate preventive strategies against the potential emergence of human pathogenic strains. This approach could involve monitoring these genomic spots and implementing measures to mitigate the risk of pathogenic gene acquisition.

## Materials and methods

### Data collection

A total of 16,506 *Salmonella* genomes were downloaded from the PathoSystems Resource Integration Center (PATRIC) [82] and NCBI genome databases in May 2021. Duplicate entries from both databases were removed. The completeness of the genome assemblies was assessed using BUSCO [83], and strains with a recommended quality score of 95 or higher were selected. The ANI scores were calculated using the tool OrthoANI [84] by comparing with the type strains *S. bongori* NCTC 12419, *S. enterica* subsp. *enterica* NCTC 12416, *S. enterica* subsp. *arizonae* NCTC 8297, *S. enterica* subsp. *diarizonae* NCTC 10060, *S. enterica* subsp. *salamae* NCTC 5773, *S. enterica* subsp. *indica* NCTC 12420, and *S. enterica* subsp. *houtenae* NCTC1 2418. Strains with an ANI score of 95% or higher were retained [84]. After applying these filters, the final dataset consisted of 12,244 genomes. The MASH tool [85,86] was used to calculate the mash distance between these strains, with a threshold of 0.1. The serovar identification and country of isolation for each strain were obtained from the information available in the PATRIC and NCBI databases. The specificity of the species, subspecies, or serovars for warm-blooded and cold-blooded hosts was determined using literature [87–90].

### Phylogeny building and pangenome analysis

The genomes were clustered using the K-mer–based tool PopPUNK v2.5.0 [91]. The model for *Salmonella* was fitted using dbscan and the phylogeny was visualised using Microreact [92] and iTOL [93]. The pangenome analysis was performed using PPanGGOLin v1.2.74 [46], and the pangenome graph was visualised using Gephi software (https://gephi.org) with the ForceAtlas2 algorithm. The RGP and the spots of insertion were extracted using the panRGP [47] subcommand of PPanGGOLin. RGP without a corresponding spot were excluded from further analysis. These included RGP on a contig border (i.e., likely incomplete) and instances in which the RGP is an entire contig (e.g., a plasmid, a region flanked with repeat sequences, or a contaminant). The frequency of the spot border gene and genes belonging to RGP were calculated using custom Python scripts (see Data Availability).

### Genome annotation and detection of genes of interest

The genomes were annotated using Prokka v1.14.6 [94]. The virulence factors, antibiotic resistance, and stress resistance genes were identified using Abricate v1.0.1 (https://github.com/tseemann/abricate) against the Comprehensive Antibiotic Resistance Database (CARD, downloaded in November 2023) [95], NCBI AMRFinderPlus (downloaded in May 2022) [96], and Virulence Factor Database (VFDB, downloaded in May 2022) [97]. The defence systems in the genomes were identified using PADLOC v1.1.0 [98] and DefenseFinder v1.0.9 [99]. The genes classified as adaptation or other categories were removed. Duplicate hits with the same gene name and location were removed using custom Python scripts (see Data Availability). Quorum-sensing genes were detected using the automatic annotation process of QSP v1.0 (https://github.com/chunxiao-dcx/QSAP) from the QS-related protein (QSP) database [100]. The virulence factors, antibiotic resistance genes, stress resistance genes, and defence systems were considered to be part of a particular RGP if the entire system was within the RGP.

### Detection of plasmid, prophage, and mobilome

The tool Plasmer [101], with default settings, was used to detect and annotate plasmids in the assemblies. Plasmid PubMLST [102] was used for plasmid typing to determine the incompatibility groups. The *Salmonella* plasmid virulence (spv) region was identified by referencing the VFDB database and mapped onto the plasmid contigs to identify pSV. Prophage regions were identified using Phigaro v2.2.6 on default mode and PhageBoost v0.1.3 [103] with a score >0.7 and a subsequent filtering with Phager (https://phager.ku.dk). From phages identified, duplicates were removed using Dedupe (https://github.com/dedupeio/dedupe) with a minimum identity of 100% and clustered at 95% identity across the region. taxmyPHAGE (https://github.com/amillard/tax_myPHAGE) was run on these regions to identify the phage kingdom, phylum, class, genus, species, and name. For virulence factors, antibiotic or stress resistance genes, or defence systems to be considered within the prophage, the entire gene cluster had to be located within the prophage region. Heat maps were generated using GraphPad Prism v9.2.0.

### Statistical analyses

Statistical analyses were performed using GraphPad Prism v9.2.0, employing simple linear regression and Pearson correlation analysis with a significance level set at a two-tailed *P*-value with a confidence interval of 95% for the correlation between the count of plasmid and antibiotic resistance genes.

## Supporting information

**S1 Fig. Distribution of virulence factors, antibiotic resistance (ABR) genes, stress resistance genes, and defence systems across the *Salmonella* genus.** The prevalence of the specific gene cluster in chromosome (black), prophage (white), or plasmid (grey) is shown as a bar graph. The data underlying this figure can be found in S7 Table.
(PDF)

**S2 Fig. Distribution of pathogenicity determinants on prophage and plasmid classes. (A)** Prevalence of virulence factors, antibiotic resistance (ABR) genes, stress resistance genes, and defence systems on different prophage genera. **(B)** Prevalence of virulence factors, ABR genes, stress resistance genes, and defence systems on different plasmid incompatibility groups. For (A) and (B), prevalence values shown represent the relative percentages of pathogenicity factors found in prophages/plasmids, indicating the proportion of these factors present within each specific prophage/plasmid class. The data underlying this figure can be found in S2, S4, and S9 Tables.
(PDF)

**S1 Table. Features of the 12,244 *Salmonella* genomes analysed in this study.**
(XLSX)

**S2 Table. Features of the plasmids found across *Salmonella* species, subspecies, and serovars.**
(XLSX)

**S3 Table. Prevalence of plasmid incompatibility groups across *Salmonella* species, subspecies, and serovars.**
(XLSX)

**S4 Table. Prevalence of prophage across *Salmonella* species, subspecies, and serovars.**
(XLSX)

**S5 Table. Features of the prophages found across *Salmonella* species, subspecies, and serovar.**
(XLSX)

**S6 Table. List of pathogenicity genes analysed in this study.**
(XLSX)

**S7 Table. Distribution and location of virulence factors, antibiotic resistance genes, stress resistance genes, and defence systems in *Salmonella* species, subspecies, and serovars, and their prevalence is plasmids and prophages.**
(XLSB)

**S8 Table. Average of plasmids, prophages, virulence factors, antibiotic resistance genes, stress resistance genes, and defence systems in *Salmonella* species, subspecies, and serovars.**
(XLSX)

**S9 Table. Prevalence (%) of virulence factors, antibiotic resistance genes, stress resistance genes, and defence systems across plasmid incompatibility groups and prophage genera.**
(XLSX)

**S10 Table. Distribution of antibiotic resistance in 12,244 *Salmonella* strains, obtained from the CARD database.**
(XLSX)

**S11 Table. Core, persistent, shell and cloud gene families identified in *Salmonella*, and their predicted function.**
(XLSX)

**S12 Table. Regions of genomic plasticity (RGP) identified in *Salmonella* species, subspecies, and serovars.**
(XLSB)

**S13 Table. RGP families and gene count in *Salmonella* spots.**
(XLSX)

**S14 Table. Virulence factors, antibiotic resistance genes, stress resistance genes, and defence systems present on spots of integration in *Salmonella*.**
(XLSB)

**S15 Table. Flanking genes defining the integration spot and their predicted function.**
(XLSX)

**S1 File. *Salmonella* phylogenetic tree in Newick format.**
(NWK)

## Acknowledgments

We acknowledge the use of the IRIDIS High-Performance Computing Facility at the University of Southampton. Infrastructure at the Center for Evolutionary Hologenomics was funded by the Danish National Research Foundation grant DNRF143.

## Author Contributions

**Conceptualization:** Franklin L. Nobrega.

**Data curation:** Simran Krishnakant Kushwaha, Franklin L. Nobrega.

**Formal analysis:** Simran Krishnakant Kushwaha, Andrew Millard, Sandhya Amol Marathe, Franklin L. Nobrega.

**Funding acquisition:** Simran Krishnakant Kushwaha, Franklin L. Nobrega.

**Investigation:** Simran Krishnakant Kushwaha, Yi Wu, Hugo Leonardo Avila, Abhirath Anand, Andrew Millard.

**Methodology:** Simran Krishnakant Kushwaha, Franklin L. Nobrega.

**Software:** Simran Krishnakant Kushwaha, Yi Wu, Hugo Leonardo Avila, Abhirath Anand, Thomas Sicheritz-Pontén.

**Supervision:** Franklin L. Nobrega.

**Writing – original draft:** Simran Krishnakant Kushwaha, Franklin L. Nobrega.

**Writing – review & editing:** Simran Krishnakant Kushwaha, Yi Wu, Hugo Leonardo Avila, Abhirath Anand, Thomas Sicheritz-Pontén, Andrew Millard, Sandhya Amol Marathe, Franklin L. Nobrega.

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
