## [Editor Report · Decision Letter 0]

26 Jan 2024

Dear Dr Nobrega, 

Thank you for submitting your manuscript entitled "Genomic plasticity is a blueprint of diversity in Salmonella lineages" for consideration as a Research Article by PLOS Biology. Again, an apology for taking so long in giving you a decision.

Your manuscript has now been evaluated by the PLOS Biology editorial staff, as well as by an academic editor with relevant expertise, and I am writing to let you know that we would like to send your submission out for external peer review.

Once your full submission is complete, your paper will undergo a series of checks in preparation for peer review. After your manuscript has passed the checks it will be sent out for review. To provide the metadata for your submission, please Login to Editorial Manager (https://www.editorialmanager.com/pbiology) within two working days, i.e. by Jan 28 2024 11:59PM.

Kind regards,

Melissa

Melissa Vazquez Hernandez, Ph.D.

Associate Editor

PLOS Biology

---

## [Decision Letter · Decision Letter 1]

18 Mar 2024

Dear Franklin,

Thank you for your patience while your manuscript "Genomic plasticity is a blueprint of diversity in Salmonella lineages" was peer-reviewed at PLOS Biology. Please accept my sincere apologies for the delays that you have experienced during the peer review process. I am currently handling your manuscript whilst my colleague Melissa is out of the office this week. Your manuscript has now been evaluated by the PLOS Biology editors, an Academic Editor with relevant expertise, and by three independent reviewers. 

In light of the reviews, which you will find at the end of this email, we would like to invite you to revise the work to thoroughly address the reviewers' reports.

As you will see, the reviewers are generally positive about your manuscript, but they raise overlapping concerns that mutational-driven resistances are not addressed in the study. After discussions with the Academic Editor, we agree that the study should be strengthened by analysing resistance by genomic mutations and this could be achieved by using antibiotic resistance databases (such as CARD) to detect such mutations in the Salmonella genome. In addition, Reviewer #1 notes that the Platon prediction tool used is now outdated since numerous tools that display higher accuracy have now been developed. Whilst re-running the whole analysis may be a significant undertaking, we ask that the revised version at least includes a confirmation for a few findings with a more up-to-date tool and discusses the limitations of using the current prediction tool. 

Given the extent of revision needed, we cannot make a decision about publication until we have seen the revised manuscript and your response to the reviewers' comments. Your revised manuscript is likely to be sent for further evaluation by all or a subset of the reviewers.

**IMPORTANT - SUBMITTING YOUR REVISION**

*Re-submission Checklist*

*Published Peer Review*

*PLOS Data Policy*

*Blot and Gel Data Policy*

Sincerely,

Richard

Richard Hodge, PhD

rhodge@plos.org

On behalf of:

Melissa Vazquez Hernandez, Ph.D

Associate Editor, PLOS Biology

REVIEWS:

Reviewer #1: The manuscript provides an overview of mobile elements and significant functional genes in Salmonella, exploring genetic diversities of plasmids, prophages, virulence genes, ARGs, and stress response genes. While offering substantial data, it lacks detailed analysis.

Several pieces of information presented in the manuscript are already well-established. For instance, the distribution of most virulence genes in the chromosome and antibiotic resistance genes in plasmids is widely known. Notably, a majority of Salmonella's virulence genes have been previously identified within at least 20 pathogenic islands (SPIs). Although the manuscript discusses SPI-1 and SPI-2, it overlooks others. These SPIs are pivotal as they demonstrate the clustering of virulence genes in the chromosome, akin to the RGPs outlined in the manuscript. It is unclear why these important previous researches had been ignored completely.

Additionally, the authors utilized Platon for plasmid prediction. However, this area is dynamic, with numerous new prediction tools emerging in the past 3-4 years, some of which appear to outperform Platon, as evidenced by benchmark comparisons (for instance, see https://journals.asm.org/doi/10.1128/spectrum.04645-22).

Concerning antibiotic resistances, the authors neglected to address mutation-driven resistance, particularly quinolone resistance resulting from mutations in QRDR. This form of resistance is particularly worrisome in Salmonella and is prevalent across various serovars, including Paratyphi A, despite the manuscript's description of this serovar as having a "minimal presence of ABR genes."

Reviewer #2: The paper is well written and describes a broad study of Salmonella genomes in the context of the prevalence of gene clusters relating to virulence, antibiotic resistance, stress and anti-phage defence systems. The authors observe cases where specific regions of the genome play host to specific gene clusters and give examples of functional interplay between the chromosome and the gene cluster. The data presented suggest that most gene clusters are not found in spots, and this angle could be explored in greater depth in the manuscript, to suggest why some gene clusters are spot-specific and others are more generalist.

The presence/absence of determinants is the main thrust of this work, which is of key importance and is well documented. However, it is also the case that mutations will impact upon virulence and antimicrobial resistance, and this is not discussed. The presence of a determinant does not necessarily equate to a functional protein or even an expressed gene, a nuance that is extremely relevant to "shaping the pathogenicity, adaptation and distribution of Salmonella". This becomes a further issue when discussing multidrug resistance (e.g. line 195-197). As written, this suggests that only serovars Indiana and Rissen harbour multidrug resistance. As mutational resistance is not considered, this view of MDR is only a partial one, which should be acknowledged.

Additional points are covered below:

1. Line 80-82. Pearce et al (citation 85) have proposed an updated range of Salmonella species and subspecies, suggest to rephrase to reflect this.

2. Line 85. Whilst Typhimurium is a main causative agent of gastroenteritis, suggesting that all 2170 strains are from human gastroenteritis is a vast oversimplification - there will be representation from invasive disease as well, and animal disease. The methods (line 550-3) state that host specificity was designated based upon literature review. Was strain specific metadata associated strains in PATRIC and NCBI used to help with this designation? Were entire serovars designated with a single host specificity (as suggested in Fig 1A)? If so, again this is an oversimplification for serovars like Typhimurium and Enteritidis.

3. Line 101-105. There is a distinction to be made here between contig number and identified type. Could a higher contig number mean a larger plasmid, rather than more plasmids?

4. Line 145. Why is it curious that SPI-1 is prevalent across all Salmonella? This has been reported previously and its presence indicates it is ancestral to the genus.

5. Line 149-153. Enteritidis is known to harbour a different virulence plasmid to Gallinarum/Pullorum (https://doi.org/10.1073/pnas.1416707112), which the rck evidence supports. 

6. Line 205-207. There is an overreaching assertion here that this study shows that the antibiotic resistance observed in S. enterica subsp. enterica is due to human antibiotic usage. Whilst this is a reasonable hypothesis, this work does not provide evidence for this assertion (e.g. that the resistance determinants identified correlate in some way with human antibiotic usage more than e.g. animal husbandry or food production) and it should be rephrased to make it clear this is a hypothesis.

7. Similar to point 6, line 236-238 makes a comment about animal production settings. Is there associated metadata with the strains to suggest that mentioned "specific serovars" are associated with animal production?

8. Line 301-304. Please clarify what "the genes" refer to. This sentence refers to Figure 3A, the legend of which refers to gene families. Which is it? Figure 3B refers to genes again.

9. Section starting line 294. The term 'persistent' has specific biological meaning, and has been applied to describe Salmonella strains that persist e.g. in the presence of antibiotic selective pressure. To clarify the use of 'persistent genes' here, it would be good to have a clear explanation of this term in the context of gene variation.

10. Lines 351-3 and 356-7. In the cases where spot presence does not correlate to expected resistance determinants, is this because the spot is occupied with alternative determinants or remains unoccupied? Is there potential for exclusion of expected determinants if alternative ones are present?

11. Line 373-378, Figure 4B and line 476-8. The figure indicates that most gene clusters are found outside of spots, and in a broader sense, only a handful of gene clusters are specifically associated with spots. This is really interesting to see, that not all gene clusters behave the same way, as some have a much greater spot specificity than others. This should be presented in greater detail - which gene functions are non-spot specific, and what might that say about their biology?

12. Line 465/141. I couldn't find the accession number for the S. bongori strain that harbours SPI-2 on NCBI - what does this number refer to?

13. Line 535. Typo - potential

14. Line 561-2. RGP without a corresponding spot were excluded from the analysis - what proportion of the data did this represent? And if it was a substantial amount, how might this affect the conclusions?

15. Line 571. Please include the versions used for these databases

Figures

1. Figure 1C. Does this actually show the average content, or a visual representation of the range of content per subspecies/serovar?

2. Figure 1E. Does the y-axis represent plasmid contigs or actual plasmid number? Please clarify what 'plasmid number' means.

3. Figure 2A. Why not show the range (like Fig 1)? How uniform are the serovars within themselves?

4. Figure 4A. What do the light grey genes represent?

Reviewer #3: This is an important and innovative analysis of the genomic landscape of the Salmonella genus. The authors have used a new approach to define "regions of genomic plasticity" that have been missed by previous investigations. The study should make an important contribution, not only to the field of Salmonella research, but more broadly to investigations of the genomic landscape of other bacterial genera. However, the manuscript is not yet ready for publication and requires a series of important modifications which are detailed below.

The key findings are based on the concept of genomic plasticity, which will be new to many readers. The authors have cited a paper from 2008 when RGP was first mentioned in the text (Mathee etal 2008, PNAS). However, since 2008, the term RGP has not been widely adopted. Consequently, at Line 49, when the authors first introduce "regions of genomic plasticity" (RGP), they need to explain precisely what the term means. As I understand it, RGP refers to specific contiguous regions of the bacterial genome that are more prone to genetic changes, rearrangements or variations? And the RGP exhibit a higher degree of flexibility and adaptability than the more conserved and stable regions of the genome?

Much of the text could be written in a clearer and less ambiguous style. For example, Lines 373 - 375 reads:

"On a broader scale, we also observe a general inclination for gene clusters with related functions to cluster within the same spots". Many phrases need to be less vague and written with more clarity.

Another example is at Line 154: "the majority of virulence factors are in chromosomal regions". Clarify, to explain that the majority of genes that encode virulence factors are located on the chromosome, and are not carried by plasmids. Make this statement less vague by stating the percentage of virulence factors that are chromosomally-encoded.

The terms prevalence, average, and frequency are central to the findings, but these terms are used interchangeably. In fact, these terms mean different things. Ensure that each term is used correctly throughout the manuscript text and the figures.

Specific Comments:

Line 22: The Abstract should be rewritten to reflect the key messages of the paper. It should be less generic and vague..

Line 82; The grouping of serovars into "host-specific, host adapted and broad-host range" should be further described (and discussed). This reviewer does not know a reliable source for this information, it is important to specify the basis of this

---

## [Editor Report · Decision Letter 2]

28 Jun 2024

Dear Dr Nobrega,

Thank you for your patience while we considered your revised manuscript "Genomic plasticity is a blueprint of diversity in Salmonella lineages" for publication as a Research Article at PLOS Biology. This revised version of your manuscript has been evaluated by the PLOS Biology editors, the Academic Editor.

Based on our Academic Editor's assessment of your revision, we are likely to accept this manuscript for publication. Please also make sure to address the following data and other policy-related requests.

a) We routinely suggest changes to titles to ensure maximum accessibility for a broad, non-specialist readership, and to ensure they reflect the contents of the paper. Please ensure you change both the manuscript file and the online submission system, as they need to match for final acceptance. In this case, we would like to suggest several possible titles, which you can choose according to what describes the study the best:

"Comprehensive blueprint of Salmonella genomic diversity identifies regions favoured by pathogenicity genes"

OR

"Comprehensive blueprint of Salmonella genomic diversity identifies hotspots for pathogenicity genes"

OR

"Comprehensive blueprint of the pathogenicity potential of Salmonella lineages"

b) Please provide a blurb between 30-40 words long. It should generally contain two sentences: the first describing the problem, and the second highlighting the study's findings.

c) We noticed that your Supplementary Table file contains all the Supplementary Tables. Please provide them as separate files, so one file per table.

Please supply the numerical values either in the a supplementary file or as a permanent DOI’d deposition for the following figures:

Figure 1ACDEF, 2ABCD, 3ABDE, 4AC, S1, S2AB

e) Please cite the location of the data clearly in all relevant main and supplementary Figure legends, e.g. “The data underlying this Figure can be found in S1 Data” or “The data underlying this Figure can be found in https://doi.org/10.5281/zenodo.XXXXX”

f) We require the tree file for Figure 1B

g) Many thanks for providing the underlying code in GitHub. However, because Github depositions can be readily changed or deleted, please make a permanent DOI’d copy (e.g. in Zenodo) and provide this URL in the manuscript and Data Availability Statement.

h) Please ensure that your Data Statement in the submission system accurately describes where your data can be found and is in final format, as it will be published as written there.

We expect to receive your revised manuscript within two weeks. 

*Published Peer Review History*

*Press*

Sincerely,

Melissa

Melissa Vazquez Hernandez, Ph.D.

Associate Editor

PLOS Biology

---

## [Editor Report · Decision Letter 3]

10 Jul 2024

Dear Franklin,

Thank you for the submission of your revised Research Article "Comprehensive blueprint of Salmonella genomic plasticity identifies hotspots for pathogenicity genes" for publication in PLOS Biology. On behalf of my colleagues and the Academic Editor, CsabaPál, I am pleased to say that we can in principle accept your manuscript for publication, provided you address any remaining formatting and reporting issues. These will be detailed in an email you should receive within 2-3 business days from our colleagues in the journal operations team; no action is required from you until then. Please note that we will not be able to formally accept your manuscript and schedule it for publication until you have completed any requested changes.

IMPORTANT: Please remember to update the abstract in the final version. I have asked my colleagues to include this request alongside their own.

PRESS

Sincerely, 

Melissa

Melissa Vazquez Hernandez, Ph.D., Ph.D.

Associate Editor

PLOS Biology
